# Identification of Epigenetic Mechanisms Involved in the Anti-Asthmatic Effects of *Descurainia sophia* Seed Extract Based on a Multi-Omics Approach

**DOI:** 10.3390/molecules23112879

**Published:** 2018-11-05

**Authors:** Su-Jin Baek, Jin Mi Chun, Tae-Wook Kang, Yun-Soo Seo, Sung-Bae Kim, Boseok Seong, Yunji Jang, Ga-Hee Shin, Chul Kim

**Affiliations:** 1Bioinformatics Group, R&D Center, Insilicogen Corporation, 35, Techno 9-ro, Yuseong-gu, Daejeon 34027, Korea; sjbaek@insilicogen.com (S.-J.B.); twkang@insilicogen.com (T.-W.K.); ghshin@insilicogen.com (G.-H.S.); 2Herbal Medicine Research Division, Korea Institute of Oriental Medicine, 1672 Yuseong-daero, Yuseong-gu, Daejeon 34054, Korea; jmchun@kiom.re.kr (J.M.C.); sys0109@kiom.re.kr (Y.-S.S.); suaa10@kiom.re.kr (S.-B.K.); 3Future Medicine Division, Korea Institute of Oriental Medicine, 1672 Yuseong-daero, Yuseong-gu, Daejeon 34054, Korea; sungbosal@kiom.re.kr (B.S.); jangbing@kiom.re.kr (Y.J.)

**Keywords:** *Descurainia sophia* (*D. sophia*) seeds, ovalbumin-induced mouse, asthma, DNA methylation

## Abstract

Asthma, a heterogeneous disease of the airways, is common around the world, but little is known about the molecular mechanisms underlying the interactions between DNA methylation and gene expression in relation to this disease. The seeds of *Descurainia sophia* are traditionally used to treat coughs, asthma and edema, but their effects on asthma have not been investigated by multi-omics analysis. We undertook this study to assess the epigenetic effects of ethanol extract of *D. sophia* seeds (DSE) in an ovalbumin (OVA)-induced mouse model of asthma. We profiled genome-wide DNA methylation by Methyl-seq and characterized the transcriptome by RNA-seq in mouse lung tissue under three conditions: saline control, OVA-induced, and DSE-treated. In total, 1995 differentially methylated regions (DMRs) were identified in association with anti-asthmatic effects, most in promoter and coding regions. Among them, 25 DMRs were negatively correlated with the expression of the corresponding 18 genes. These genes were related to development of the lung, respiratory tube and respiratory system. Our findings provide insights into the anti-asthmatic effects of *D. sophia* seeds and reveal the epigenetic targets of anti-inflammatory processes in mice.

## 1. Introduction

Asthma is a chronic inflammatory and multi-causal disease determined by various genetic, epigenetic and environmental factors [1,2]. In recent years, epigenetic marks have emerged as a potential mechanism for asthma [3,4,5,6], and have been shown to regulate many immune cell processes involved in the disease. In addition, several epigenomics and transcriptomics studies have identified DNA methylation as a biomarker for asthma [7,8,9].

Dietary phytochemicals are not only a rich source of minerals, vitamins and micronutrients, but also contain bioactive components such as phenols, phenolic acids, flavonoids and flavonoid glycosides. These bioactive compounds exhibit great potential against many diseases [10,11], including asthma [12]. *Descurainia sophia* Webb ex Prantl, which contains several such bioactive compounds [13,14], is one of the original species of *Descurainia Sophia* (L.) Webb ex Prantl [15]. The seeds of this plant have been used traditionally to treat diseases and symptoms such as cough, asthma and edema [16,17].

Although *D. sophia* seeds have not been extensively investigated, recent studies have revealed the major active components and their effects. For instance, ethanol extract and volatile oil of *D. sophia* seeds inhibit the growth of various cancer cell lines in vitro [16,18,19]. Several constituents isolated from these seeds exert cytotoxic and anti-inflammatory effects on both human cancer cell lines and murine macrophages [17]. Based on their potential therapeutic effects, these components are predicted to be useful in the treatment of allergies and inflammatory lung diseases. Although the anti-asthmatic effects of *D. sophia* seeds have been partially characterized, to date no study has taken an integrated multi-omics approach to investigate the therapeutic mechanisms underlying their target pathways in asthma. To obtain insight into the epigenetic mechanisms of the anti-asthmatic effects of *D. sophia* seed extract (DSE) in an ovalbumin (OVA)-induced mouse model of asthma, we performed Methyl-seq and RNA-seq to profile genome-wide DNA methylation and gene expression, respectively. In addition, we performed an integrated analysis using epigenomic and transcriptomic data sets (Appendix A). The resultant integrated network of DNA methylation and expression revealed epigenetically regulated genes associated with anti-asthmatic effects.

## 2. Results

### 2.1. DSE Treatment Reduces Asthmatic Inflammation in an OVA-Induced Mouse Model

We collected samples from mice subjected to three treatments: saline (control, *n* = 3), OVA (asthma-induced mice; *n* = 3) and DSE (herbal treatment; *n* = 4) (Appendix A). To evaluate the anti-asthmatic effects of DSE, we monitored the phenotypes of allergic lung inflammation, including histopathological features, the number of infiltrated cells and cytokine expression. First, to evaluate the effects of DSE on lung inflammation in asthma, we sectioned the lungs and stained the sections with hematoxylin–eosin (H&E). Infiltration of immune cells around blood vessels was higher in OVA-induced mice than in saline-treated mice, but lower in DSE-treated mice (Figure 1A). Next, to confirm that DSE inhibits inflammatory cell infiltration, we counted inflammatory cells, including eosinophils, neutrophils and macrophages, in bronchoalveolar lavage fluid (BALF). In OVA-induced mice, both total inflammatory cells and individual categories of cells (eosinophils, neutrophils and macrophages) were more abundant than in the saline control group, whereas DSE-treated mice had significantly fewer inflammatory cells than untreated asthmatic mice (Figure 1B–E). Furthermore, the levels of interleukin (IL)-4 were significantly elevated in asthmatic mice, but significantly reduced in DSE-treated mice (Figure 1F). *IL-4*, a type 2 cytokine, plays a key role in the pathogenesis of allergic asthma [20], and *IL-4* levels are a major marker of type 2 helper T cells and allergic inflammation. These results confirmed that DSE inhibited OVA-induced allergic lung inflammation by decreasing inflammatory cell infiltration and production of the Th2 cytokine *IL-4* in the lung.

### 2.2. Methyl-Seq Reveals Distinctive DNA Methylation Changes among Saline-Treated, OVA-Induced and DSE-Treated Mice

We performed DNA methylome profiling in samples from the same animals to identify differentially methylated regions (DMRs) among the three groups. For all CpGs with a minimum read depth 20, as determined by Methyl-seq, we observed a bimodal distribution of DNA methylation (Appendix A). We then performed principal component analysis (PCA) on the genome-wide methylation dataset, as shown in Figure 2A. Approximately 5.2% (23,154 bins) of hyper methylated and 2.75% (12,247 bins) of hypo methylated DMRs (Diff ≥ 10% and *q*-value < 0.01) were selected from a total of 445,288 bins in OVA-induced samples vs. controls (Figure 2B and Appendix A), whereas 1.07% (4818 bins) of hyper methylated and 0.97% (4379 bins) of hypo methylated DMRs were selected from a total of 449,539 bins in DSE-treated vs. OVA-induced samples (Figure 2C and Appendix A).

Overall, DMRs tended to be detected in functional regions of the genome, such as promoters and coding regions. In OVA-induced samples vs. controls, hyper methylated regions were more frequent than hypo methylated regions in promoter, exon, intron and intergenic regions. By contrast, in the comparison of DSE-treated vs. OVA-induced samples, DMRs did not differ significantly between hyper and hypo methylated regions, except in promoter regions (Figure 2D).

We focused on methylation patterns that differed between saline-treated and OVA-induced mice, and were restored to the control level by DSE treatment, i.e., hyper methylated in OVA vs. saline and hypo methylated in DSE vs. OVA (“hyper/hypo”; Figure 3A and Appendix A) or hypo methylated in OVA vs. saline and hyper methylated in DSE vs. OVA (“hypo/hyper”; Figure 3B and Appendix A). These DMRs, which were identified as key genes, were used in the next step to infer the mechanism underlying the effects of the herbal treatment. We then used the DAVID resource to examine the functional annotation of genes (Figure 3C,D). Genes with the hyper/hypo pattern shown in Figure 3A were enriched in KEGG pathway such as olfactory transduction (*p*-value: 5.11 × 10^−23^; Calm1, Cnga3, Prkacb, and Prkg1), cytokine–cytokine receptor interaction (*p*-value: 1.3 × 10^−8^; *Hgf*, *Flt*, *Prlr*, *Tgfbr1*, *Kit*, *Cntfr*, *Acvrl1*, *Egfr*, *Il2Rb*, *Ifnar2*, *Crlf2*, and *Il7*), and chemokine signaling pathway (*p*-value: 2.3 × 10^−7^; *Adcy2*, *Prkacb*, *Raf1*, *Prkcz*, *Pik3ca*, *Elmo1*, and *Pik3r1*; Figure 3C and Appendix A). Conversely, genes with the hypo/hyper pattern shown in Figure 3B were enriched in functions related to cytokine–cytokine receptor interaction (*p*-value: 9.58 × 10^−7^; *Csf3r*, *Vegfa*, *Ccl28*, *Il7r*, *Ifng*, and *Crlf2*) and natural killer cell mediated cytotoxicity (*p*-value: 1.2 × 10^−4^; *Nfat5* and *Ifng*; Figure 3D and Appendix A).

### 2.3. Identification by Multi-Omics Analysis of Epigenetically Regulated Genes that Were Restored to Control Levels by DSE Treatment

To identify genes regulated by DNA methylation in anti-asthmatic processes, we examined the RNA-seq data for negative correlations between DNA methylation and gene expression. We characterized two distinct patterns: Pattern A, including 17 genes that were downregulated in OVA vs. saline and upregulated in DSE vs. OVA (“down/up”) and hyper/hypo methylated (Figure 4A and Table 1), and pattern B, consisting of only a single gene that was upregulated in OVA vs. saline and downregulated in DSE vs. OVA (“up/down”) and hypo/hyper methylated (Figure 4B and Table 1). Both patterns exhibited a negative correlation between DNA methylation and gene expression (Appendix A). In functional terms, these 18 genes were enriched in GO biological process terms pertaining to development of the lung alveolus (*p*-value: 0.001; *Vegfa*, *Hopx*, and *Errfi1*), lung vasculature (*p*-value: 0.007; *Vegfa* and *Errfi1*), and lung (*p*-value: 0.017; *Vegfa*, *Hopx*, and *Errfi1*; Appendix A).

### 2.4. Identification of Two Main Anti-Asthma Genes by Integrated Network Analysis

To understand the interactions among the anti-asthmatic genes, we performed integrated network analysis of DNA methylation and gene expression during recovery by DSE treatment. Using the 18 genes detected in the pattern analysis described above, we examined the interactions using Ingenuity Pathway Analysis (IPA) software and found 60 network-eligible genes and chemicals associated with asthma (Appendix A). To determine the effects of methylation changes, we overlaid the DNA methylation values on the network. Figure 5 shows the predicted pathways related to anti-asthma genes in OVA vs. saline and DSE vs. OVA. In OVA vs. saline, *Vegfa* was upregulated, whereas *Kit*, *Acsl4*, *Hopx*, *Rps24*, *Errfi1*, *Gtf2e2*, *Ints6*, *2310035C23Rik* (KIAA1468) and *Sppl3* were downregulated. Methylation changes were in the opposite direction from expression level changes, as shown in Figure 5A (bar plots next to genes).

Figure 5B shows the predicted pathways in DSE vs. OVA. The majority of genes, including *Kit*, *Acsl4*, *Hopx*, *Rps24*, *Errfi1*, *Gtf2e2*, *Ints6*, *2310035C23Rik* (KIAA1468) and *Sppl3*, were upregulated; only *Vegfa* was downregulated; and the methylation changes were the opposite of those observed in OVA vs. saline. Two hub genes (*Vegfa* and *Kit*) in these integrated networks exhibited an inverse correlation between DNA methylation and gene expression. Hub genes, which are highly interconnected nodes in IPA modules, are functionally significant. Thus, *Vegfa* (vascular endothelial growth factor A) and *Kit* (proto-oncogene receptor tyrosine kinase) were identified as the main regulators of the integrated network, potentially explaining the differential regulation of their downstream target genes among the control, OVA-induced and DSE-treated groups.

## 3. Discussion

The aim of this study was to discover distinct differential methylomic signatures in lung tissue samples obtained from an asthma-induced mouse model. The results of H&E staining and BALF analysis confirmed that DSE inhibited allergic lung inflammation by decreasing inflammatory cell infiltration and IL-4 production (Figure 2). Therefore, we investigated the possibility that DSE could be used as a treatment for asthma. To understand how asthmatic inflammation (OVA-induced) and anti-asthmatic effects (DSE-treated) regulate respiratory disease, and how to determine which genes cause disease, we compared differences among saline, OVA-induced and DSE-treated mice by RNA-seq and Methyl-seq. Previous studies that investigated the epigenetic mechanisms underlying asthma [3,21] and characterized the anti-asthmatic effects [22,23,24,25,26] partly explained the epigenetic regulation of anti-asthmatic effects by herbal medicine. However, the results of our study provide an integrated perspective on the epigenetic changes involved in the anti-asthmatic effects of *D. sophia* seeds, as determined by genome-wide gene expression and methylation profiling. This is the first integrated study of the role of DNA methylation and gene expression in the anti-asthmatic effects of *D. sophia* seeds. Our findings effectively identify key genes underlying anti-asthmatic patterns and elucidate an integrated network related to asthma.

To identify anti-asthmatic changes in methylation, we profiled genome-wide methylation patterns, including the hyper/hypo and hypo/hyper patterns described in the Results section, during OVA induction followed by treatment with a medicinal herb. Gene Ontology (GO) analysis of these candidate genes revealed strong enrichment of terms related to olfactory disorders, myeloid cells, osteoclasts and cytokine/chemokine signaling pathways. Although asthma has various causes, airway inflammation is a common triggering mechanism [27,28,29,30]. Airway inflammation is mediated by chemokines and cytokines [31,32,33], suggesting that changes in the methylation of genes associated with cytokines/chemokines play an important role in the onset of asthma. Additionally, we also observed enrichment in ECM–receptor interaction and neurotrophin signaling (Appendix A). ECM-receptors have direct and indirect effects on cell regulation and interact with eosinophilic asthma [10,34], whereas growth factors such as neurotrophins and their receptors are important in normal lung development, developmental lung disease, allergies and inflammatory conditions such as rhinitis or asthma [35,36,37,38,39].

By combining Methyl-seq with RNA-seq, we identified 18 candidate genes for which DNA methylation changed as a result of the anti-asthmatic effects of DSE. GO analysis revealed that genes whose expression returned to control levels upon DSE treatment were enriched in asthma-related functions such as lung alveolus development, lung vasculature development, and vasculature development.

In addition, we identified highly connected hub genes in integrated networks of DNA methylation and gene expression. Notably, *Vegfa* was regulated in an “up/down” manner, due to its hypo/hyper methylation pattern, in OVA-induced vs. saline and DSE-treated vs. OVA-induced mice. *Vegfa*, a member of the vascular endothelial growth factor family, promotes allergic inflammation and plays an important role in Th2 inflammation [40]. Furthermore, several studies have reported that *Vegfa* levels are increased in tissues and biological samples from asthma patients [40,41,42]. *Kit*, which was regulated in a “down/up” fashion, due to its hyper/hypo methylation pattern, is related to the NF-κB signaling pathway, which regulates pro-inflammatory mediators [43], and may help to maintain alveolar structure [44]. These results indicate that methylation changes of two hub genes play important roles in regulating the anti-inflammatory effect. We also identify previously unreported genes with concordant molecular changes in asthma: *Shroom2*, *Pcmtd1*, *Zfp568* and *2310035C23Rik* were significantly down/upregulated and hyper/hypo methylated.

This study had several limitations, including the fact that Methyl-seq technology was used to profile the methylome without any investigation of histone modifications, e.g., by ChIP-seq. In various omics analyses, it is important to keep in mind that expression of a candidate gene could be affected by methylation changes in a distal enhancer, or that a given gene could contain an enhancer region for other genes. Further work will be needed to examine changes in methylation at distal enhancer regions. Other omics studies such as single-cell or DNA resources based approaches also will be needed for identification of various causes. Moreover, larger sample sizes will be required for a more powerful characterization of the epigenetic mechanisms involved in restoration of control levels (of methylation or gene expression) by DSE treatment.

## 4. Materials and Methods

### 4.1. Plant Material and Preparation of DSE

Seeds of *D. sophia* were collected by Dr. Sungyu Yang and Dr. Byeong Cheol Moon of the Korea Institute of Oriental Medicine (KIOM) in Geumsan, Chungcheongnam-do, Korea. The plants used in this study were authenticated based on macroscopic morphological characteristics by Dr. Goya Choi of KIOM. The voucher specimens were deposited in the Korean Herbarium of Standard Herbal Resources (herbarium code KIOM, Specimen 2-18-0137) at the KIOM. Dried seeds of *D. sophia* (760 g) were extracted twice with 70% ethanol (with 3 h reflux) at 80 °C, and the extract was concentrated under reduced pressure. Yields of DSE were approximately 7.44% (*wt*/*wt*).

### 4.2. In Vivo Experiments

#### 4.2.1. Animals

Eight-week-old female BALB/c mice were purchased from Daehan BioLink Co., Ltd. (Chungcheongbuk-do, Korea) and housed under specific pathogen-free conditions with freely available food and water. All animal care and experimental procedures were performed with the approval of the Institutional Animal Care and Use Committee (IACUC) of the KIOM (17-073).

#### 4.2.2. Induction of Lung Inflammation and the Administration of Drug

Asthma was induced as previously described with some modifications [45,46]. To induce allergic lung inflammation, mice were sensitized with OVA twice, 7 days apart (i.e., on Days 0 and 7) by intraperitoneal injection of OVA (50 µg) emulsified with 200 µL of aluminum sulfate (Alum; InvivoGen, San Diego, CA, USA). Beginning on Day 12, the mice were challenged with 50 µL of OVA (25 µg) intranasally for 4 days under anesthesia with 2% isoflurane (Piramal Critical Care Inc., Bethlehem, PA, USA) delivered by a Vevo™ Compact Anesthesia System (FUJIFILM VisualSonics, Toronto, ON, Canada). The mice were divided into four groups (*n* = 7 per group). One was injected with saline only, and the remaining three groups were sensitized to OVA. On Days 9–15, the treatment groups received oral administration of vehicle, DSE (200 mg/kg) for 7 consecutive days.

#### 4.2.3. Histological Analysis

For euthanasia, an overdose of pentobarbital sodium (a short-acting barbiturate) was injected in accordance with IACUC guidelines. Lung tissues of mice were collected and fixed in 10% formalin. Fixed tissues were embedded in paraffin and cut into 5-μm sections with a microtome (Leica, Nussloch, Germany). The sections were deparaffinized and stained with H&E for analysis of inflammatory changes.

#### 4.2.4. Collection of BALF, Quantification of Inflammatory Cells and Measurement of IL-4 Levels in BALF

BALF was collected by endotracheal lavage with 1 mL of Dulbecco’s phosphate-buffered saline (ScienCell Research Laboratories, Carlsbad, CA, USA) via intubation. To determine differential cell counts in the BALF, cells were placed on a slide by cytospin and then stained with Hema 3 solution (Fisher HealthCare, Pittsburgh, PA, USA). The proportion of each cell type was determined based on morphology, and the number of cells was calculated by multiplying the proportions by the total cell count. To measure cytokine levels, BALF was centrifuged for 5 min at 4 °C, and then the levels of IL-4 were measured in the supernatant by an ELISA, as previously described [47]. Briefly, 11B11 antibody was used to capture IL-4, and BVD6-24G2 biotinylated antibody was used for its detection. Antibodies were purchased from BD Biosciences (San Diego, CA, USA).

### 4.3. Methyl-Seq and Data Analysis

SureSelect Methylation libraries were prepared according to the manufacturer’s instructions (Agilent Methyl-seq protocol). Three micrograms of high-quality genomic DNA were diluted with 50 µL EB buffer and fragmented to a median size of 150 bp using the Covaris-S2 instrument (Covaris, Woburn, MA, USA). Following successful shearing, the gDNA was subjected to end blunting, dA-tailing and the ligation with methylated adapters. The constructed libraries for the capture were quantified using the Quant-iT dsDNA HS Assay Kit (Invitrogen, Carlsbad, CA, USA) with the Invitrogen Qubit fluorometer (Invitrogen, Carlsbad, CA, USA). The hybridizations were performed using a thermal cycler that was set at 65 °C for 24 h with the lid heated to 105 °C; then, Dynabeads^®^ MyOne Streptavidin T1 (Invitrogen, Carlsbad, CA, USA) was used to capture the biotinylated RNA probes. Finally, the captured library samples were washed and eluted for the bisulfite conversion. Eluted DNA was bisulfite converted using the EZ DNA Methylation-Gold kit (ZymoResearch, Cat. #D5005, Carlsbad, CA, USA), following the Agilent instruction manual. Furthermore, the treated DNA was amplified by PCR, converting uracil residues in the sample to thymidine. Amplified bisulfied-treated libraries were purified using AMPure XP beads (Beckman Coulter, High Wycombe, UK). To index the captured libraries, bisulfied-treated libraries were amplified by PCR. Indexed libraries were purified and accessed by Agilent 2100 Bioanalyzer (High Sensitivity DNA assay, Agilent, Santa Clara, CA, USA). Sequencing runs were performed in paired-end mode using the HiSeq 2500 platforms (Illumina, San Diego, CA, USA). Using the TruSeq SBS Kits v4 (Illumina, San Diego, CA, USA) for the HiSeq, sequencing-by-synthesis reactions were extended for 101 cycles from each end.

After sequencing, the reads were trimmed to remove adapters and low-quality reads (per-base quality < 20) and thereby improve paired-end mapping using Cutadapt (v. 1.13) [48]. Reads were mapped to the Mus musculus genome (mm10) with Bismark (v. 0.17.0) [49]. The methylKit R package (v. 1.6.0) [50] was used for analysis of Methyl-seq data and DMR identification (window size: 1000 and step size: 500). Regions with two or more CpGs, ≥10% mean methylation difference and *q*-value < 0.01 were considered significant. methylKit software (v. 1.9.0) was used for annotation of peaks and assessment of the distribution of methylation peaks across genomic features. Genomic features were classified into six types of regions (intergenic, 5′ UTR, 3′ UTR, promoter, CDS and intron) based on the UCSC genome annotation.

### 4.4. RNA-Seq and Data Analysis

Total RNA was also extracted from lung tissues used in the methyl-seq using Nucleo spin RNA kit (MACHREY-NAGEL, Dueren, Germany). The quantity and quality of the total RNA were evaluated using Agilent 2100 Bioanalyzer (Agilent, Santa Clara, CA, USA) and assessing the RNA quality indicator (RIN). Library preparation was performed with an Illumina TruSeq Stranded mRNA sample preparation kit (Illumina, San Diego, CA, USA) according to the manufacturer′s instructions. Indexing adapters were ligated to the ends of the cDNA fragments using ligation mix reagent at 30 °C for 10 min. After twice washing with sample purification bead, PCR was performed to enrich those cDNA fragments that have adapter molecules on both ends. Thermocycler conditions were as follows: 98 °C for 30 s; 15 cycles of 98 °C for 10 s, 60 °C for 30 s, and 72 °C for 30 s; and a final extension at 72 °C for 5 min. Finally, quality and band size of libraries were assessed using Agilent 2100 Bioanalyzer. Libraries were quantified by qPCR using CFX96 Real Time System (Biorad, Hercules, CA, USA). After normalization, sequencing of the prepared library was conducted on the Nextseq 500 (Illumina, San Diego, CA, USA) with 76 bp paired-end reads.

Reads were trimmed to remove adapters and low-quality reads (per-base quality < 20) and thereby improve paired-end mapping. High-quality sequence reads were mapped to the Mus musculus genome (mm10) using Hisat2 (v. 2.1.0) [51], and gene expression levels were quantified with the ballgown R package (v. 2.12.0) [52]. Genes differentially expressed among the three groups were evaluated by the edgeR package (v. 3.0.8) [53], which is based on negative binomial models for RNA-seq count data. Differentially expressed genes were screened with a cutoff threshold of log (FC) ≥ |1| and *p*-value < 0.05.

### 4.5. Correlation Analysis between DNA Methylation and Expression

To identify genes with an inverse relationship between expression and methylation, the expression level was adjusted to a range from 0 to 1, as for methylation, and then Spearman rank correlations were used to assess the relationship between CpG methylation and gene expression. To find the functional mechanisms involving these genes, we conducted GO term and KEGG pathway analysis using DAVID (v. 6.7) [54], which provides a set of data-mining tools that systematically combine functionally descriptive data with intuitive graphical displays.

### 4.6. Pathway Analysis

IPA (v. 1.13, Qiagen, Valencia, CA, USA) software was used to predict the networks containing the 18 candidate genes whose expression changed due to alterations in DNA methylation in response to anti-asthmatic treatment. First, we used network connection tools in IPA to display known functional interactions between these genes, and then further integrated our findings by overlaying genes involved in asthma, lung development and immune disease. To determine how each gene affects the integrated network, we overlaid the fold change in gene expression and DNA methylation ∆β values. In the figures, bar plots beside the genes in the network show DNA methylation and expression change under each condition. Molecules are represented as nodes, and the biological relationships between nodes are represented as edges (line). The intensity of node color indicates the degree of expression change.

### 4.7. Data Access

The data generated over the course of this study have been deposited into the Gene Expression Omnibus (GEO) under accession number GSE114587 (https://www.ncbi.nlm.nih.gov/geo/query/acc.cgi?acc=GSE114587; secure token for reviewer: itytgkuoptclnej).

## 5. Conclusions

In summary, the integrative approach described here identified anti-asthmatic changes in a mouse model, identifying 18 genes and integrated networks involved in methylation and gene expression. We identified several epigenetically regulated genes and pathways involved in anti-asthmatic effects, which should facilitate development of novel and effective therapeutic agents for asthma.

## Figures and Tables

**Figure 1 molecules-23-02879-f001:**
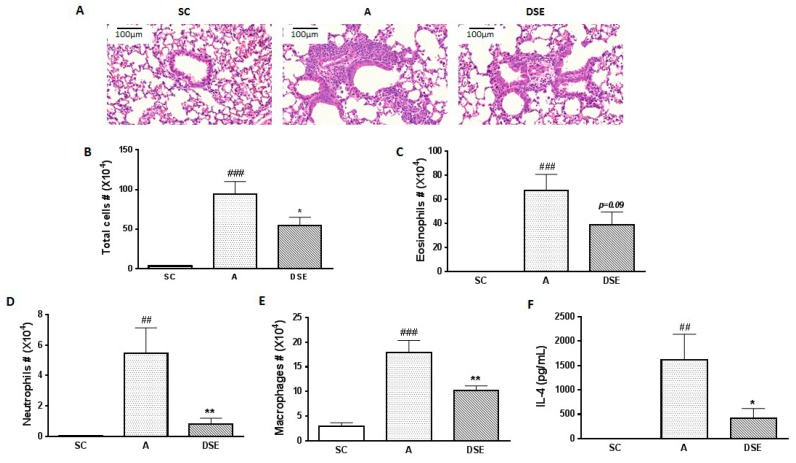
Effects of DSE on phenotypes of allergic asthma in OVA-induced mice. Representative photographs of lung sections stained with H&E (magnification, ×200) (**A**). Counts of: total cells (**B**); eosinophils (**C**); neutrophils (**D**); and macrophages (**E**) in infiltrated BALF. Levels of IL-4 in BALF determined by ELISA (**F**). Data are presented as means ± SEM (*n* = 7). ## *p* < 0.01, ### *p* < 0.001 compared with the saline control group. * *p* < 0.05, ** *p* < 0.01 in allergic lung inflammation vs. vehicle group. SC, Saline Control; A, OVA; DSE, *D. sophia* seed extract.

**Figure 2 molecules-23-02879-f002:**
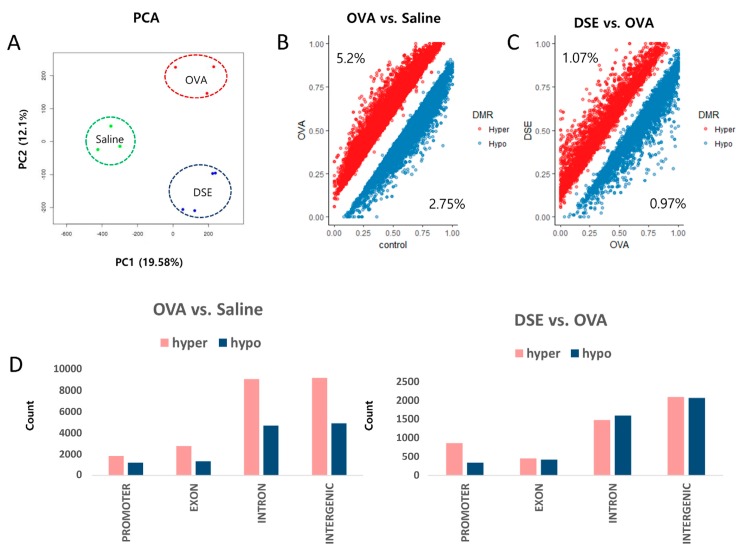
Genome-wide methylation profiling in DMRs. PCA plot of DNA methylation data. Samples are colored according to the experimental group (**A**). Scatter plot of DMRs in OVA vs. saline control (**B**). Scatter plot of DMRs in DSE vs. OVA (**C**). Distribution of DMRs in OVA vs. control (**left**), and DSE vs. OVA (**right**) (**D**).

**Figure 3 molecules-23-02879-f003:**
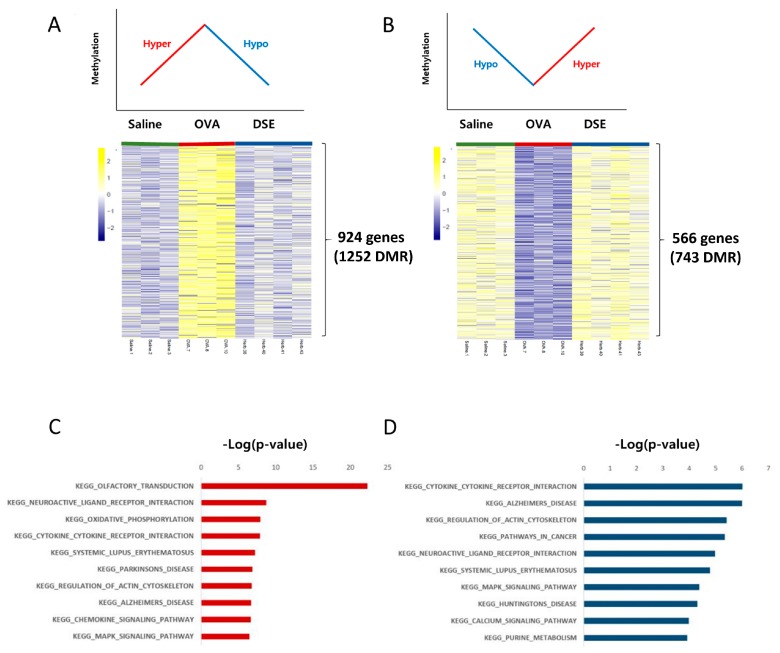
Anti-asthmatic methylation patterns in response to DSE treatment. Heat map of hyper/hypo methylated genes in OVA vs. saline and DSE vs. OVA (**A**). Heat map of hypo/hyper methylated genes under the same conditions (**B**). Top 10 functional annotation results of hyper/hypo methylation pattern, consisting of 924 genes (1252 DMRs) (**C**), and hypo/hyper methylation pattern, containing 566 genes (743 DMRs) (**D**). KEGG pathway names are shown on the left, and the bars on the right represent the −log (*p*-value) of the corresponding pathway.

**Figure 4 molecules-23-02879-f004:**
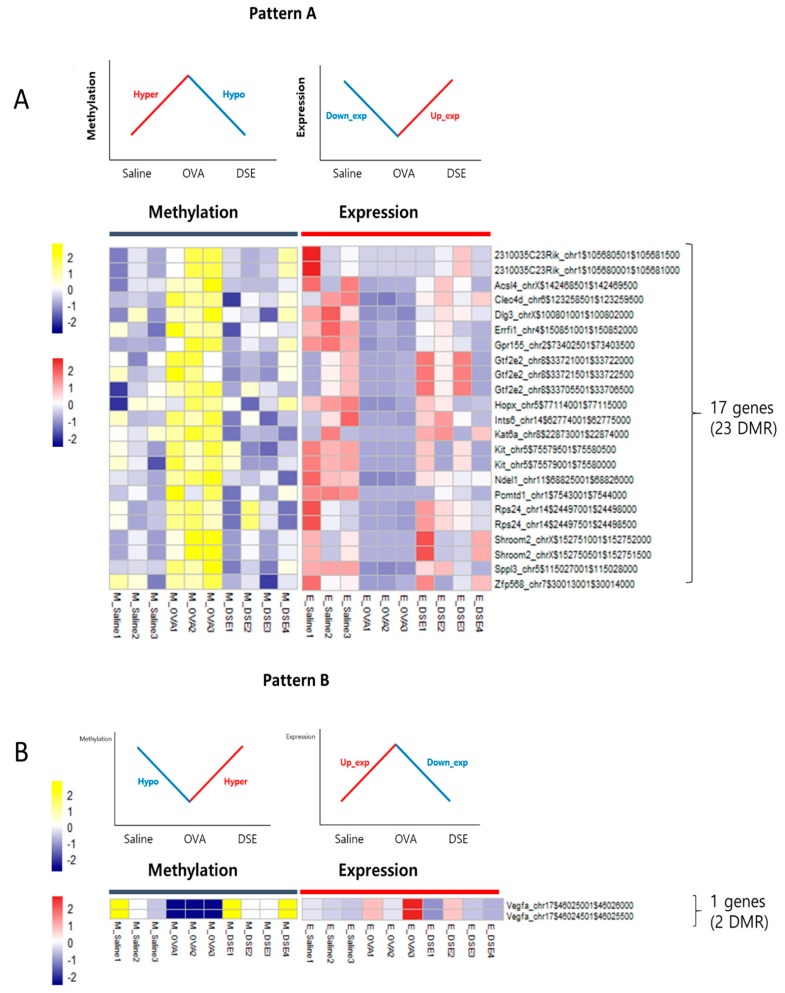
Anti-asthmatic pattern genes with inverse correlation between methylation and expression. Matched heat map of DNA methylation and gene expression profiling in hyper/hypo methylated and down/upregulated genes (**A**). Matched heat map of DNA methylation and gene expression in hypo/hyper methylated and up/downregulated genes (**B**).

**Figure 5 molecules-23-02879-f005:**
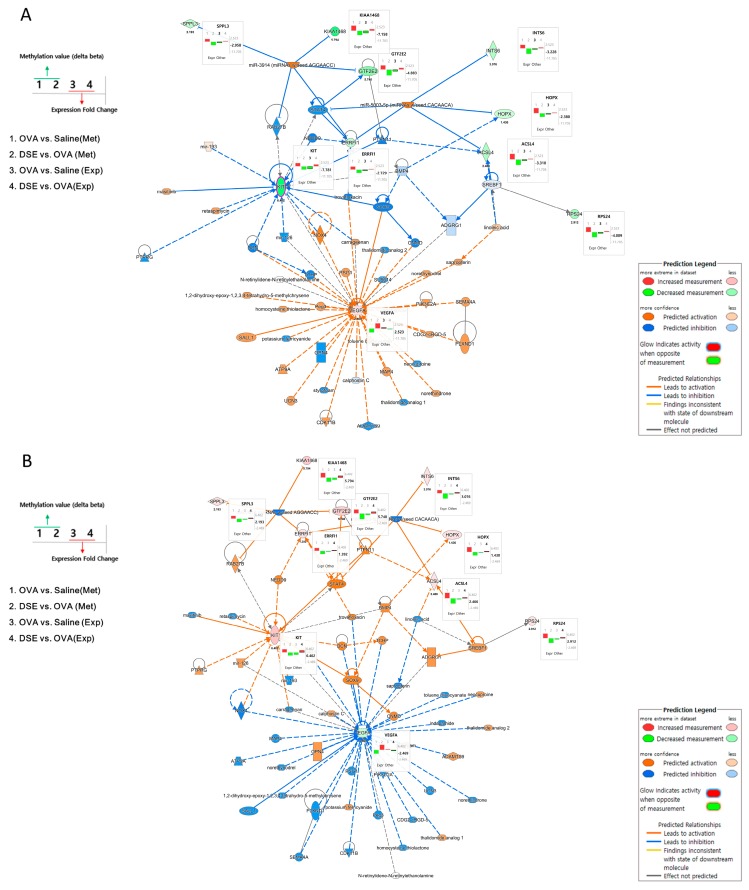
Gene networks associated with anti-asthma effects. Pathway depicts expression levels of anti-asthmatic genes regulated by DNA methylation changes in OVA vs. saline using IPA (**A**). Pathway depicts expression levels of anti-asthmatic genes regulated by DNA methylation changes in DSE vs. OVA (**B**). Bar plots next to genes in the network show changes in methylation and expression levels under each condition. The left panel of the network shows the bar plot legend, and the right panel provides a legend of the IPA network.

**Table 1 molecules-23-02879-t001:** Candidate genes of integrative pattern analysis.

Gene	Pattern	Transcript Name	DMR Position	Methylation				Expression			
				OVA vs. Saline		DSE vs. OVA		OVA vs. Saline		DSE vs. OVA	
				Delta-Beta	*p*-Value	Delta-Beta	*p*-Value	Fold Change	*p*-Value	Fold Change	*p*-Value
*2310035C23Rik*	Pattern A	NM_173187	chr1$105680001$105681000	17.91792	0.001044	−14.3229	0.000793	−7.15782	0.003233	5.794239	0.01018
*2310035C23Rik*	Pattern A	NM_173187	chr1$105680501$105681500	17.91792	0.001044	−14.6318	0.000691	−7.15782	0.003233	5.794239	0.01018
*Acsl4*	Pattern A	NM_019477	chrX$142468501$142469500	20.1579	4.01 × 10^−8^	−18.5894	3.60 × 10^−9^	−3.31049	0.008914	2.466142	0.036953
*Clec4d*	Pattern A	NM_010819	chr6$123258501$123259500	13.24306	3.05 × 10^−6^	−12.7135	8.06 × 10^−8^	−1.8897	0.031682	1.659789	0.047727
*Dlg3*	Pattern A	NM_001177780	chrX$100801001$100802000	21.42624	2.03 × 10^−5^	−15.8962	0.000187	−3.39981	0.001902	2.512896	0.013947
*Errfi1*	Pattern A	NM_133753	chr4$150851001$150852000	10.49137	6.31 × 10^−5^	−10.8417	1.84 × 10^−6^	−2.72862	0.000324	1.392076	0.044534
*Gpr155*	Pattern A	NM_001190297	chr2$73402501$73403500	14.45583	0.001214	−13.4888	0.000127	−2.73567	0.000478	1.421815	0.047225
*Gtf2e2*	Pattern A	NM_026584	chr8$33705501$33706500	14.15904	2.14 × 10^−7^	−11.7841	8.08 × 10^−8^	−4.88319	0.011646	5.739732	0.003803
*Gtf2e2*	Pattern A	NM_026584	chr8$33721001$33722000	17.11279	0.000173	−19.3001	6.24 × 10^−8^	−4.88319	0.011646	5.739732	0.003803
*Gtf2e2*	Pattern A	NM_026584	chr8$33721501$33722500	14.40352	0.000466	−20.3312	8.98 × 10^−12^	−4.88319	0.011646	5.739732	0.003803
*Hopx*	Pattern A	NM_001159900	chr5$77114001$77115000	17.89375	2.10 × 10^−9^	−11.912	5.71 × 10^−6^	−2.37969	0.00221	1.429905	0.04741
*Ints6*	Pattern A	NM_008715	chr14$62774001$62775000	12.51032	0.0003	−18.3492	3.51 × 10^−9^	−3.22793	0.001325	3.076277	0.001547
*Kat6a*	Pattern A	NM_001081149	chr8$22873001$22874000	21.42651	0.002265	−40.3226	7.75 × 10^−14^	−2.32499	0.028763	3.024011	0.004454
*Kit*	Pattern A	NM_001122733	chr5$75579001$75580000	11.51523	7.89 × 10^−5^	−10.3308	3.31 × 10^−5^	−7.78059	0.000103	6.402056	0.00061
*Kit*	Pattern A	NM_001122733	chr5$75579501$75580500	10.35753	1.89 × 10^−6^	−11.8328	2.81 × 10^−10^	−7.78059	0.000103	6.402056	0.00061
*Ndel1*	Pattern A	NM_023668	chr11$68825001$68826000	21.79487	0.000464	−22.6016	3.14 × 10^−5^	−4.91466	0.000645	4.250178	0.001895
*Pcmtd1*	Pattern A	NM_183028	chr1$7543001$7544000	10.6038	0.001635	−11.7335	2.64 × 10^−5^	−5.37129	3.98 × 10^−7^	2.821372	0.002068
*Rps24*	Pattern A	NM_011297	chr14$24497001$24498000	17.13836	0.000641	−17.2688	0.00011	−4.04976	0.001056	2.911984	0.01074
*Rps24*	Pattern A	NM_207635	chr14$24497001$24498000	17.13836	0.000641	−17.2688	0.00011	−4.00879	0.000414	3.224884	0.002518
*Rps24*	Pattern A	NM_011297	chr14$24497501$24498500	17.13836	0.000641	−17.2688	0.00011	−4.04976	0.001056	2.911984	0.01074
*Rps24*	Pattern A	NM_207635	chr14$24497501$24498500	17.13836	0.000641	−17.2688	0.00011	-4.00879	0.000414	3.224884	0.002518
*Shroom2*	Pattern A	NM_001290684	chrX$152750501$152751500	19.51472	1.06 × 10^−5^	−16.7372	4.44 × 10^−6^	−11.7053	2.06 × 10^−6^	9.44746	0.00004
*Shroom2*	Pattern A	NM_001290684	chrX$152751001$152752000	19.51472	1.06 × 10^−5^	−16.7372	4.44 × 10^−6^	−11.7053	2.06× 10^−6^	9.44746	0.00004
*Sppl3*	Pattern A	NM_029012	chr5$115027001$115028000	10.11934	9.88 × 10^−7^	−11.4787	2.84 × 10^−11^	−2.95759	0.000555	2.19341	0.006261
*Zfp568*	Pattern A	NM_001167873	chr7$30013001$30014000	10.90426	0.00247	−17.1518	1.59 × 10^−7^	−4.33247	0.007645	4.172162	0.008244
*Vegfa*	Pattern B	NM_001025257	chr17$46024501$46025500	−13.0325	0.004499	18.71345	3.94 × 10^−6^	2.522877	0.014275	−2.46869	0.007728
*Vegfa*	Pattern B	NM_001025257	chr17$46025001$46026000	−13.0325	0.004499	18.71345	3.94 × 10^−6^	2.522877	0.014275	−2.46869	0.007728

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
