# Peer review of "Identification of Epigenetic Mechanisms Involved in the Anti-Asthmatic Effects of Descurainia sophia Seed Extract Based on a Multi-Omics Approach"

_molecules, 2018, doi:10.3390/molecules23112879_

Round 1
Reviewer 1 Report
This study approached the epigenetic effects of the ethanol extract of D. sophia seeds (DSE, 200 mg/kg for 7 consecutive days) in the lung tissue of an ovalbumin (OVA)-induced mouse model of asthma. The results showed that DSE treatment reduces asthmatic inflammation and this asthmatic effect is associated with the changes of the DNA methylation and gene expression of 18 genes. Further integrated network analysis Identified two main anti-asthma genes, Vegfa (vascular endothelial growth factor A) and Kit (proto-oncogene receptor tyrosine kinase), as the main regulators of the integrated network. There are some minor concerns for the present manuscript as listed in the following:
(1) The method of lung tissue preparation for methyl-seq and RNA-seq was not described.
(2) It is better to describe why the dose (200 mg/kg) for DSE is selected.
(3) [P2: We collected samples from mice subjected to three treatments: saline (control, n=3), OVA (asthma-induced mice; n=3) and DSE (herbal treatment; n=4) (Figure S2)] vs. [P3: Fig.1- Data are presented as means ± SEM (n = 7)] vs. [P10: The mice were divided into four groups (n = 7 per group).]??
(4) Typos and others:
P1: in the disease4.
P1: such bioactive compound
P3: Fig.1- ***p < 0.005?
P4: cytokine cytokine receptor interaction
P5: Vegfa was
P10: oral administration of vehicle, DSE (200 mg/kg) or 7 consecutive days.
P12: References-check all the inconsistent format for the cited paper title (capital letter-R11, R13 and others), journal name (abbreviation-R13), and pages (R13: 1379-1390; R14: 917-925; R26: 303-311)
Author Response
Response to Reviewer 1 Comments
Comments and Suggestions for Authors:
This study approached the epigenetic effects of the ethanol extract of D. sophia seeds (DSE, 200 mg/kg for 7 consecutive days) in the lung tissue of an ovalbumin (OVA)-induced mouse model of asthma. The results showed that DSE treatment reduces asthmatic inflammation and this asthmatic effect is associated with the changes of the DNA methylation and gene expression of 18 genes. Further integrated network analysis Identified two main anti-asthma genes, Vegfa (vascular endothelial growth factor A) and Kit (proto-oncogene receptor tyrosine kinase), as the main regulators of the integrated network. There are some minor concerns for the present manuscript as listed in the following:
Point 1: The method of lung tissue preparation for methyl-seq and RNA-seq was not described.
Response 1: Thank you for valuable comment.
We fully agree with this comment and following your recommendations we have added the method of lung tissue preparation for methyl-seq and RNA-seq.
Detailed modifications of manuscript are below:
Modified manuscript –Page 12~13 Material and Methods section (present submission)
Point 2: It is better to describe why the dose (200 mg/kg) for DSE is selected.
Response 2: Thank you for valuable comment.
To determine dose DSE, we considered two aspects; our preliminary results and articles on the efficacy of animal experiments on DSE.
To determine and compare the anti-inflammatory effects of plant species (200 mg/kg) on allergic lung inflammation, using OVA-induced airway inflammation in mice. As a results, among them, the excellent effect of DSE (200 mg/kg) was confirmed and further experiments were conducted by selecting this material (reviewer’s Fig. 1).
In addition, some previous studies used from 200 mg/kg to 400 mg/kg as oral doses1).
Therefore, based on the above study, we used the dose levels of 200 mg/kg as effective doses.
Reviewer’s Fig. 1. Effect of five species on total, eosinophila and differential cell counts in the bronchoalveolar lavage fluid of asthmatic mice. The mouse groups are as follows: SC, saline-injected control group; A, ovalbumin treatment alone; DDN, JS, KDJ, KDD, SB; mice treated with ovalbumin and DDN, JS, KDJ, KDD, SB extract (200 mg/kg); DEX; mice treated with ovalbumin and Dexamethasone (5 mg/kg). The data are presented as the mean ± standard error of the mean (n=4~6). ###p < 0.001 vs. Saline; *p < 0.05 vs. OVA, **p < 0.01 vs. OVA, ***p < 0.001 vs. OVA.
DDN; Lepidium apetalum JS; Descurainia sophia KDJ; Draba nemorosa KDD; Lepidium virginicum SB; Erysimum cheiranthoides
Point 3: [P2: We collected samples from mice subjected to three treatments: saline (control, n=3), OVA (asthma-induced mice; n=3) and DSE (herbal treatment; n=4) (Figure S2)] vs. [P3: Fig.1- Data are presented as means ± SEM (n = 7)] vs. [P10: The mice were divided into four groups (n = 7 per group).]??
Response 3: Thank you for valuable comment.
To evaluate the anti-asthmatic effects of DSE, we performed at first, efficacy verification of DSE on ovalbumin (OVA)-induced mouse model of asthma (four groups of seven mice in each group). The next, for epigenetic mechanisms of DSE, we collected samples in OVA induced mice (four groups of three to four mice in each group). For this reason, the number of mice was shown for each experiment.
Point 4: Typos and others:
Response 4: Thank you for valuable comment.
We have revised the manuscript accordingly, and detailed corrections are listed below point by point:
Point 4.1: in the disease4.
Response 4.1:
Changed ‘in the disease4’ to ‘in the disease’ (Page 1-present submission).
Point 4.2: such bioactive compound
Response 4.2:
Changed ‘such bioactive compound’ to ‘such bioactive compounds’ (Page 1-present submission).
Point 4.3: Fig.1- ***p < 0.005?
Response 4.3:
Changed the sentence by eliminating “***p < 0.005” as it was not needed (Page 3; Fig 1-present submission).
Point 4.4: cytokine cytokine receptor interaction
Response 4.4:
Cytokine-cytokine receptor interaction - Mus musculus (mouse) (KEGG pathway ID: PATHWAY: mmu04060)
Changed ‘cytokine cytokine receptor interaction’ to ‘cytokine-cytokine receptor interaction’ (Page 4-present submission).
Point 4.5: Vegfa was
Response 4.5:
Changed ‘Vegfa’ to ‘Vegfa’ (Page 6-present submission).
Point 4.6: oral administration of vehicle, DSE (200 mg/kg) or 7 consecutive days.
Response 4.6:
Changed ‘DSE (200 mg/kg) or 7 consecutive days’ to ‘DSE (200 mg/kg) for 7 consecutive days’ (P11 –present submission).
Point 4.7: References-check all the inconsistent format for the cited paper title (capital letter-R11, R13 and others), journal name (abbreviation-R13), and pages (R13: 1379-1390; R14: 917-925; R26: 303-311)
Response 4.7:
A suggested by reviewer, we revised references in presented manuscript accordance with consistent format for the cited paper title, journal name, and pages.
Changed upper case to lower case (R2, R11, R13, R16, and R26)
Changed journal name (Journal of the American Oil Chemists` Society) to abbreviation of journal name (J Am Oil Chem Soc).
Changed pages (R13:1379-1390; R14:917-925; R26:303-311) to new page (R13:1379-90; R14:917-25; R26:303-11).

Reviewer 2 Report
An interesting study on a medical problem rapidly expanding around the world, therefore, the search for pharmacologically useful substances and the knowledge of their mechanism of action is very useful. The authors investigated the molecular mechanisms that characterize the use of Descurainia sophia seeds that are traditionally used to treat coughs, but also for asthma and oedema.
The anti-asthmatic effect of the alcoholic extract of these seeds was first ascertained in the model animals for asthma (mice) used for testing by means of histological tests, a specific biochemical test (IL4) and the counts of inflammatory cells in the BALF. Authors have performed Methyl-seq, as well as RNA-seq, in order to sketch differential genome-wide DNA methylation and gene expression in the control and treated groups of mice.
While I agree on the experimental design implemented by the authors and on the approach used to analyze their results, I have however a reservation about the meaning of their DNA methylome profiling.
Epigenetics discusses about chemical modifications of DNA, by thinking that these modifications can influence gene activity and expression, but without changes in DNA sequence. Fundamentally, the issue is that this change should reflect a change within cells. But cells show subtype compositions or heterogeneities or variability of DNA sequences, and so on, all factors that may affect the cell state. Sometimes differences in DNA methylation when comparing various sets of samples, may show confounding influences like the proportions of cell subtypes, DNA sequence polymorphism, often not accounted for.
What did the authors to highlight these effects? Have they evaluated the potential error? Indeed the changes shown may depend also on intrinsic causes of the system studied.
However, in any case, a word of caution should be given to readers on this issue. This referee is well aware that a control of these effects requires studies that are more detailed than those performed by authors. I mean that not all the changes that are measured could be epigenetic in nature, so you have to get your hands on and explain something.
Author Response
Response to Reviewer 2 Comments
Point 1: An interesting study on a medical problem rapidly expanding around the world, therefore, the search for pharmacologically useful substances and the knowledge of their mechanism of action is very useful. The authors investigated the molecular mechanisms that characterize the use of Descurainia sophia seeds that are traditionally used to treat coughs, but also for asthma and oedema.
The anti-asthmatic effect of the alcoholic extract of these seeds was first ascertained in the model animals for asthma (mice) used for testing by means of histological tests, a specific biochemical test (IL4) and the counts of inflammatory cells in the BALF. Authors have performed Methyl-seq, as well as RNA-seq, in order to sketch differential genome-wide DNA methylation and gene expression in the control and treated groups of mice.
While I agree on the experimental design implemented by the authors and on the approach used to analyze their results, I have however a reservation about the meaning of their DNA methylome profiling.
Epigenetics discusses about chemical modifications of DNA, by thinking that these modifications can influence gene activity and expression, but without changes in DNA sequence. Fundamentally, the issue is that this change should reflect a change within cells. But cells show subtype compositions or heterogeneities or variability of DNA sequences, and so on, all factors that may affect the cell state. Sometimes differences in DNA methylation when comparing various sets of samples, may show confounding influences like the proportions of cell subtypes, DNA sequence polymorphism, often not accounted for.
What did the authors to highlight these effects? Have they evaluated the potential error? Indeed the changes shown may depend also on intrinsic causes of the system studied.
However, in any case, a word of caution should be given to readers on this issue. This referee is well aware that a control of these effects requires studies that are more detailed than those performed by authors. I mean that not all the changes that are measured could be epigenetic in nature, so you have to get your hands on and explain something.
Response 1: Thank you for valuable comment.
We totally agree with this comment about intrinsic causes of the system studied by methylome. However, the method used in this study seems to have a limitation to identify the essential cause mentioned. We have approached the association between methylation and gene expression considering inverse correlation. In next studies, we will explore various causes in addition to the factors caused from epigenetics by considering cell type and mutation through research using single-cell1) or genomic DNA resources2). Accordingly, the following contents were added.
Added: And other omics studies like as single-cell or DNA sources based approaches also will be needed for identification of various causes. (Discussion section- page 11)
Reference;
Trapnell C., Defining cell types and states with single-cell genomics. Genome Res. 2015 Oct;25(10):1491-8. (PMID: 26430159)
Francesconi M, Lehner B,. The effects of genetic variation on gene expression dynamics during development. Nature. 2014 Jan 9;505(7482):208-11. (PMID: 24270809)
Reviewer 3 Report
I have reviewed an article: "Identification of epigenetic mechanisms involved in the anti-asthmatic effects of Descurainia sophia seed extract based on a multi-omics approach". I think this article is worth for publication, bu there are some issues that I would like to ask you to explain in more detail and put it into a paper;
1) Did you run chemical analysis of D. sophie extracts? Which compound(s) are taught to be responsible for anti asthmatic effect?
2) Are these compounds stable during metabolism and they reach lungs in native form or they are metabolized so metabolites have this anti asthmatic effect?
3) Are there a differences in metabolic pathways in mice and humans considering metabolism of D. sophia seed extracts?
Author Response
Response to Reviewer 3 Comments
Comments and Suggestions for Authors:
I have reviewed an article: "Identification of epigenetic mechanisms involved in the anti-asthmatic effects of Descurainia sophia seed extract based on a multi-omics approach". I think this article is worth for publication, but there are some issues that I would like to ask you to explain in more detail and put it into a paper;
Point 1: Did you run chemical analysis of D. sophie extracts? Which compound(s) are taught to be responsible for anti asthmatic effect?
Response 1: Thank you for valuable comment.
To identify and quantify the levels of marker components in DSE, Ultra-high performance liquid chromatography (UHPLC) analysis was performed. The chromatogram of the main components is shown in reviewer’s Fig. 2. The five components of DSE were linolenic acid (0.055 ± 0.01%), linoleic acid (0.068 ± 0.01%), palmitic acid (0.042 ± 0.01%), oleic acid (0.038 ± 0.00%), and stearic acid (0.006 ± 0.00%). Our chemical analysis results of DSE is part of other papers and should only be viewed as a reviewer’s figure.
Our results have confirmed the major fatty acid components of DSE, and some studies have also shown that essential fatty acids are essential for the development and degradation of inflammatory pathways associated with asthma pathophysiology1). Therefore, the effects of asthma on the components of DSE need further study.
Reference;
1) Wendell SG et al. Fatty acids, inflammation, and asthma. Allergy Clin Immunol. 2014 May;133(5):1255-64.
Reviewer’s Fig. 2. Chromatogram of standard (A) and Descurainia sophia extracts (B) detected by charged aerosol detector; 1. Linolenic acid, 2. Linoleic acid, 3. Palmitic acid, 4. Oleic acid, 5. Stearic acid.
Point 2: Are these compounds stable during metabolism and they reach lungs in native form or they are metabolized so metabolites have this anti asthmatic effect?
Response 2: Thank you for valuable comment.
Based on the chemical analysis of reports1) and our results, we found that DSE contains the most of essential oils and fatty acids. These fatty acids have been reported to modulate prostaglandin metabolism and to exhibit anti-inflammatory activities by suppressing leukotriene B4 generated by arachidonic acid in linoleic acid2).
These previous studies suggest that PUFAs (linolenic acid, linoleic acid, and oleic acid) have a beneficial relationship with lung function, and asthma prevalence, and therefore, the anti-asthmatic effect on the active compounds of each of the DSE constituted mainly of fatty acids should also be studied.
With respect to fatty acid metabolites and anti-asthmatic effects, several reports have indicated that omega-3 fatty acids or their metabolites function as protective molecules in murine models of asthma3). These bioactive metabolites showed their preventive effects on airway eosinophilic inflammation, airway hyperresponsiveness and inflammatory cytokine production in OVA-induced asthmatic responses4, 5). These results suggest that the fatty acids of DSE and their metabolites are expected to have anti-asthmatic effects.
Reference;
1) Gong JH et al., Extractions of oil from Descurainia sophia seed using supercritical CO2, chemical compositions by GC-MS and evaluation of the anti-tussive, expectorant and anti-asthmatic activities., Molecules. 2015 Jul 22;20(7):13296-312.
2) Simopoulos AP., Omega-3 fatty acids in inflammation and autoimmune diseases., J Am Coll Nutr. 2002 Dec;21(6):495-505.
3) Miyata J et al., Role of omega-3 fatty acids and their metabolites in asthma and allergic diseases., Allergol Int. 2015 Jan;64(1):27-34.
4) Morin C et al., Docosahexaenoic acid derivative prevents inflammation and hyperreactivity in lung: implication of PKC-Potentiated inhibitory protein for heterotrimeric myosin light chain phosphatase of 17 kD in asthma., Am J Respir Cell Mol Biol. 2011 Aug;45(2):366-75.
5) Morin C et al., MAG-EPA resolves lung inflammation in an allergic model of asthma.Clin Exp Allergy. 2013 Sep;43(9):1071
Point 3: Are there a differences in metabolic pathways in mice and humans considering metabolism of D. sophia seed extracts?
Response 3: Thank you for valuable comment.
In previous studies have shown that species differences in metabolic capacity between human and mouse livers1). They have shown that the activities of several enzymes (lipase, UGT, ADH, ALDH, and so on) involved in the metabolism of di (2-ethylhexyl) phthalate (DEHP) in human and mouse livers. And they noted that lipase activity significantly influenced the internal amount of mono (2-ethylhexyl) phthalate (MEHP)2, 3). We already described above, fatty acids are essential for development and degradation of inflammatory pathways considering metabolism of D. Sophia seeds extract. Therefore further study will be need to investigated difference in metabolic pathway in mice and human considering metabolism of D. Sophia seed extract (ex.lipase).
Reference;
1) Ito Y et al., Species and inter-individual differences in metabolic capacity of di(2-ethylhexyl) phthalate (DEHP) between human and mouse livers., Environ Health Prev Med. 2014 Mar;19(2):117-25.
2) Ito Y et al., Species differences in the metabolism of di(2-ethylhexyl) phthalate (DEHP) in several organs of mice, rats, and marmosets., Arch Toxicol. 2005 Mar;79(3):147-54.
3) Ito Y et al., Induction of peroxisome proliferator-activated receptor alpha (PPARalpha)-related enzymes by di(2-ethylhexyl) phthalate (DEHP) treatment in mice and rats, but not marmosets., Arch Toxicol. 2007 Mar;81(3):219-26.
In this study, we focused on “epigenetic regulation” of anti-asthmatic effects of Descurainia sophia seeds. Like as the reviewer's valuable comment (point 1, 2 3), we are currently working on various compounds that affect the anti-asthmatic effect of D. Sophia seed extract. Thank you for valuable suggestions!!
